

# Analysis of prognostic genes in the tumor microenvironment of lung adenocarcinoma

Zhan-yu Xu[*], Mengli Zhao[*], Wenjie Chen, Kun Li, Fanglu Qin, Wei-wei Xiang, Yu Sun, Jiangbo Wei, Li-qiang Yuan, Shi-kang Li and Sheng-hua Lin

Department of Thoracic and Cardiovascular Surgery, The First Affiliated Hospital of Guangxi Medical University, Nanning, China

[*] These authors contributed equally to this work.

Corresponding authors
Shi-kang Li, 201013064@qq.com, lsksci10@163.com
Sheng-hua Lin, weipeng@stu.gxmu.edu.cn, 450443222@qq.com

## ABSTRACT

**Background**. Prognostic genes in the tumor microenvironment play an important role in immune biological processes and the response of cancer to immunotherapy. Thus, we aimed to assess new biomarkers that are associated with immune/stromal cells in lung adenocarcinomas (LUAD) using the ESTIMATE algorithm, which also significantly affects the prognosis of cancer.

**Methods**. The RNA sequencing (RNA-Seq) and clinical data of LUAD were downloaded from the the Cancer Genome Atlas (TCGA ). The immune and stromal scores were calculated for each sample using the ESTIMATE algorithm. The LUAD gene chip expression profile data and the clinical data (GSE37745, GSE11969, and GSE50081) were downloaded from the Gene Expression Omnibus (GEO) for subsequent validation analysis. Differentially expressed genes were calculated between high and low score groups. Univariate Cox regression analysis was performed on differentially expressed genes (DEGs) between the two groups to obtain initial prognosis genes. These were verified by three independent LUAD cohorts from the GEO database. Multivariate Cox regression was used to identify overall survival-related DEGs. UALCAN and the Human Protein Atlas were used to analyze the mRNA /protein expression levels of the target genes. Immune cell infiltration was evaluated using the Tumor Immune Estimation Resource (TIMER) and CIBERSORT methods, and stromal cell infiltration was assessed using xCell.

**Results**. In this study, immune scores and stromal scores are significantly associated with the clinical characteristics of LUAD, including T stage, M stage, pathological stage, and overall survival time. 530 DEGs (18 upregulated and 512 downregulated) were found to coexist in the difference analysis with the immune scores and stromal scores subgroup. Univariate Cox regression analysis showed that 286 of the 530 DEGs were survival-related genes ($p < 0.05$). Of the 286 genes initially identified, nine prognosis-related genes (CSF2RB, ITK, FLT3, CD79A, CCR4, CCR6, DOK2, AMPD1, and IGJ) were validated from three separate LUAD cohorts. In addition, functional analysis of DEGs also showed that various immunoregulatory molecular pathways, including regulation of immune response and the chemokine signaling pathways, were involved. Five genes (CCR6, ITK, CCR4, DOK2, and AMPD1) were identified as independent prognostic indicators of LUAD in specific data sets. The relationship between the expression levels of these genes and immune genes was assessed. We found that CCR6

mRNA and protein expression levels of LUAD were greater than in normal tissues. We evaluated the infiltration of immune cells and stromal cells in groups with high and low levels of expression of CCR6 in the TCGA LUAD cohort. In summary, we found a series of prognosis-related genes that were associated with the LUAD tumor microenvironment.

## INTRODUCTION

Lung cancer is a malignant disease with the highest morbidity and mortality worldwide in recent years (*Bray et al., 2018*). Eighty-five percent of lung cancers are non-small cell lung cancer (NSCLC), which can be divided into lung adenocarcinomas (LUAD) and squamous carcinomas (*Liu et al., 2017a*). There has been an increase in the incidence of LUADs, accounting for 60% of NSCLCs and the most common type of lung cancer (*Denisenko, Budkevich & Zhivotovsky, 2018*). The 5-year survival rate of patients with early-stage NSCLC after surgical treatment is 70%–90%; however, approximately 75% of patients lost their opportunity for surgery by the time of initial diagnosis (*Goldstraw et al., 2016*). Despite important advances in targeted molecular therapies and immunotherapy for lung cancer, drug resistance still limits success (*Ascierto et al., 2019*), and new biomarkers are warranted for different populations. This will have important implications for improving patients' early diagnosis rates and for discovering novel targeted therapies.

In addition to tumor cells, stromal cells and immune cells as well as tumor-associated normal epithelial cells form malignant solid tumor tissues (*Moffitt et al., 2015*). Stromal cells contain cancer-related fibroblasts, pericytes, mesenchymal stem cells, blood, and lymphatic endothelial cells (*Quail & Joyce, 2013*). In the tumor microenvironment (TME), stromal cell components and tumor cells interact with other to establish a dynamic connection, which affects the biological function and chemical resistance of tumor cells (*Joyce & Pollard, 2009*). Four types of immune cells are relevant: tumor infiltrating lymphocytes (TILs), tumor-associated macrophages, myeloid-derived suppressor cells, and dendritic cells (*Bedognetti et al., 2019*). TILs play an important role in immune surveillance and killing of cancerous cells in patients with lung cancer (*Ye et al., 2017*). NK cell-mediated tumor suppressor function is closely related to tumor progression and is a key factor regulating lung cancer growth and metastasis (*Fang, Xiao & Tian, 2017*). Depicting the immune infiltration of tumor microenvironment or determining the proportion of relevant normal cells in tumor tissues can help in building an accurate tumor prognosis and prediction model. *Guo et al. (2017)* demonstrated that chondroitin sulfate proteoglycan serglycin (SRGN), which is secreted by tumor cells and stromal components in the TME, promotes malignant phenotypes through interacting with tumor cell receptor CD44. Increased expression of SRGN is associated with poor prognosis in primary LUAD (*Guo et al., 2017*). Galectin-3 is a lectin that contributes to TME immunosuppression and regulates diverse functions. Studies suggest that increased Gal-3 expression during cancer progression

augments tumor growth, invasiveness, metastatic potential, and immune suppression (*Compagno et al., 2014*; *Farhad, Rolig & Redmond, 2018*)

The quantification of various cell types allows accurate analysis of dynamically changing immune microenvironments. Experimental methods, such as single-cell sequencing (*Kyrochristos et al., 2019*), are expensive and cumbersome to operate; therefore, various algorithms have been developed to describe tumor immune environments, including single-sample gene set enrichment analysis (GSEA) (*Foroutan et al., 2018*), MCPcounter (*Petitprez et al., 2018*), and CIBERSORT (*Chen et al., 2018*). *Yoshihara et al. (2013)* proposed a new algorithm known as ESTIMATE that uses expression data to estimate the number of immune and stromal cells in malignant tumor tissues. This algorithm focuses on the formation of immune and stromal cells from major non-tumor components of tumor samples and identifies the stromal and tumor tissue-specific characteristics associated with immune cell infiltration. Its effectiveness has been shown in glioblastoma (*Jia et al., 2018*) and breast cancer (*Winslow et al., 2016*). Although accurate immune and stromal scoring of LUAD can be performed, there have been no detailed studies that have analyzed related the expression profiles with ESTIMATE. In this study, the microenvironment-related genes of the LUAD cohort of the TCGA database were studied based on the ESTIMATE score, and the prognostic genes of three different LUAD cohorts from the GEO database were verified. To further elucidate the immunological mechanisms, we assessed the role of the immune microenvironment in the development of LUAD using immune/stromal cell infiltration analysis.

## MATERIAL AND METHODS

### Database

The level 3 gene expression profile data of lung adenocarcinoma patients (using Illumina HiSeq_RNASeqV2 lung adenocarcinoma RNA expression profiles) were downloaded from the TCGA website (https://portal.gdc.cancer.gov/). The immune and stromal scores were calculated from each sample using the ESTIMATE algorithm. In addition, patient clinical data were downloaded from the TCGA official website (https://portal.gdc.cancer.gov/repository). The LUAD gene chip expression profile data and the clinical data (GSE37745, GSE11969, and GSE50081) of the samples were downloaded from the GEO official website (https://www.ncbi.nlm.nih.gov/geo/) for subsequent validation analysis. To analyze the mRNA/protein expression of CCR6 in primary LUAD and normal tissues, UALCAN (http://ualcan.path.uab.edu) was used, which was based on the TCGA database and Clinical Proteomic Tumor Analysis Consortium. To validate the expression of CCR6, the Human Protein Atlas (http://www.proteinatlas.org/) was used. Tumor Immune Estimation Resource (TIMER) (https://cistrome.shinyapps.io/timer/) was used to assess the correlation between differentially expressed gene (DEG) expression and the immune cell infiltration level. Immune cell infiltration was evaluated using the TIMER and CIBERSORT method, and stromal cell infiltration was determined using xCell (https://xcell.ucsf.edu/).

### Identification of differentially expressed genes (DEGs)

The limma package in R was used for the analysis of differential expression genes (*Ritchie et al., 2015*), and the cutoff value was set to |log fold change(FC)| > 1, and a *p* value < 0.05. We plotted a heat map using the cluster analysis results of DEGs in R.

### GO and KEGG enrichment analysis

To investigate the molecular functions, biological processes, cell components, and signaling pathways involved in DEGs, we used the DAVID database (v6.8, https://david.ncifcrf.gov/) to perform GO analysis and KEGG pathway enrichment. *p* value < 0.05 and FDR < 0.05 were marked as significant terms.

### Statistical analysis

The data were assessed for normal distribution, and the central tendency was expressed as the mean ± standard deviation for the data with normal distribution. Means were compared using a Student's *t*-test. Continuous variables that did not have normal distribution were analyzed using the Wilcoxon rank sum test or Kruskal–Wallis rank sum test. Student's t-tests were used to compare the distribution of scores in the high and low groups, gender, and M stages. The Wilcoxon rank sum test was used to compare the distribution of scores in age, T stages, and N stages. The Kruskal–Wallis rank sum test was used to compare the differences in the distribution of scores of the four different pathological stages. Spearman's rank correlation coefficient was used to measure the correlation between lung cancer stage and overall survival. Determination of survival-related genes was performed using univariate Cox regression analysis (with a *p* value of < 0.05) and the corresponding survival curves were plotted. Multivariate Cox analysis was used to evaluate the contribution of genes as independent prognosis factors for patient survival. All tests were two-tailed, and *p* values less than 0.05 were considered statistically significant. Al analyses were performed using SPSS for Windows, software version 25.0 (SPSS Inc., Chicago, IL).

## RESULTS

### Correlation between immune and stromal scores and clinical characteristics of LUAD

We downloaded the gene expression profiles and clinical data of 492 patients with LUAD from the TCGA database, including 225 males (45.7%), 267 females (54.3%), 157 patients under 60 years of age (31.9%), and 335 patients over 60 years of age (68.1%). The ESTIMATE algorithm revealed stromal scores of −1959.31 to 2989.77, and the immune scores of −1355.85 to 2905.3. The results of the relationship between the two scores and clinical data (TNM and pathological stage) are shown in Table 1. There were differences in the distribution of stromal scores between the sexes and across M stages. Immune scores also differed with respect to T and pathological stages.

We found that the immune and stromal scores of female patients were significantly higher than those of from male patients ($p = 0.0069$ and $0.0049$, respectively) (Figs. 1A and 1B). The immune score of T1 was significantly higher than that of T2/T3/T4 stages ($p = 0.0002$, Fig. 1C); however, but the distribution of stromal scores was not different

 

**Table 1   The relationship between the two scores and clinical characteristics.**

| Variable | Patients | Stromal scores | P value | Immune scores | P value |
|---|---|---|---|---|---|
| **Scores** | | | <0.000** | | <0.000** |
| Low | 246 | −539.40 ± 27.69 | | 490.60 ± 45.44 | |
| High | 246 | 635 ± 26.75 | | 1437 ± 41.31 | |
| **Age, years** | | | 0.223 | | 0.100 |
| ≥ 60 | 157 | 235.10 | | 231.1 | |
| >60 | 335 | 251.84 | | 253.72 | |
| **Sex** | | | 0.007** | | 0.005** |
| Male | 225 | −48.34 ± 48.91 | | 849.60 ± 55.64 | |
| Female | 267 | 128.80 ± 43.52 | | 1060 ± 49.79 | |
| **T stage** | | | 0.101 | | 0.001** |
| T1 | 167 | 260.99 | | 277 | |
| T2/T3/T4 | 325 | 238.85 | | 230.40 | |
| **N stage** | | | 0.768 | | 0.155 |
| N0 | 318 | 242.34 | | 247.45 | |
| N1/N2/N3 | 163 | 238.39 | | 228.42 | |
| **M stage** | | | 0.010** | | 0.037* |
| M0 | 324 | 54.67 ± 40.45 | | 961.70 ± 46.01 | |
| M1 | 24 | −343.60 ± 137.40 | | 592.90 ± 177.2 | |
| **Stage** | | | 0.137 | | 0.023* |
| I | 262 | 249.35 | | 256.45 | |
| II | 118 | 246.36 | | 242.92 | |
| III | 79 | 232.61 | | 210.16 | |
| IV | 25 | 183.72 | | 196.48 | |

**Notes.**
*$p < 0.05$.
**$p < 0.01$.

between the two groups ($p = 0.0999$) (Fig. 1D). The immune and stromal scores of the M0 stage was higher than those of the M1 stage ($p = 0.0098$ and $0.0366$, respectively) (Figs. 1E and 1F). For the different pathological stages of LUAD, the immune scores were different at each stage ($p = 0.0276$, Fig. 1G) and gradually decreased from stage I to stage IV (Spearman correlation coefficients: $\rho = -0.131$, $p = 0.004$); however, but the difference in stromal scores among the four pathological stages was not statistically significant ($p = 0.1525$, Fig. 1H).

Next, we evaluated the correlations between immune (or stromal) scores and overall survival. LUADs ($N = 492$) were divided into high- and low-score groups, with 246 cases in each group. Stromal scores ranged from −959.31 to 36.85 for the low group and from 39.8 to 2098.77 for the high group. Immune scores ranged from −1355.85 to 952.61 for the low group and 964.93 to 2905.3 for the high group. The Kaplan–Meier survival curve analysis for immune scores indicated that the median survival of the low-score group was lower than that of the high-score group (1194 d vs. 1499 d, $p = 0.011$; log-rank test; Fig. 1I). Similarly, for stromal scores, the median survival of the low-score group was lower than

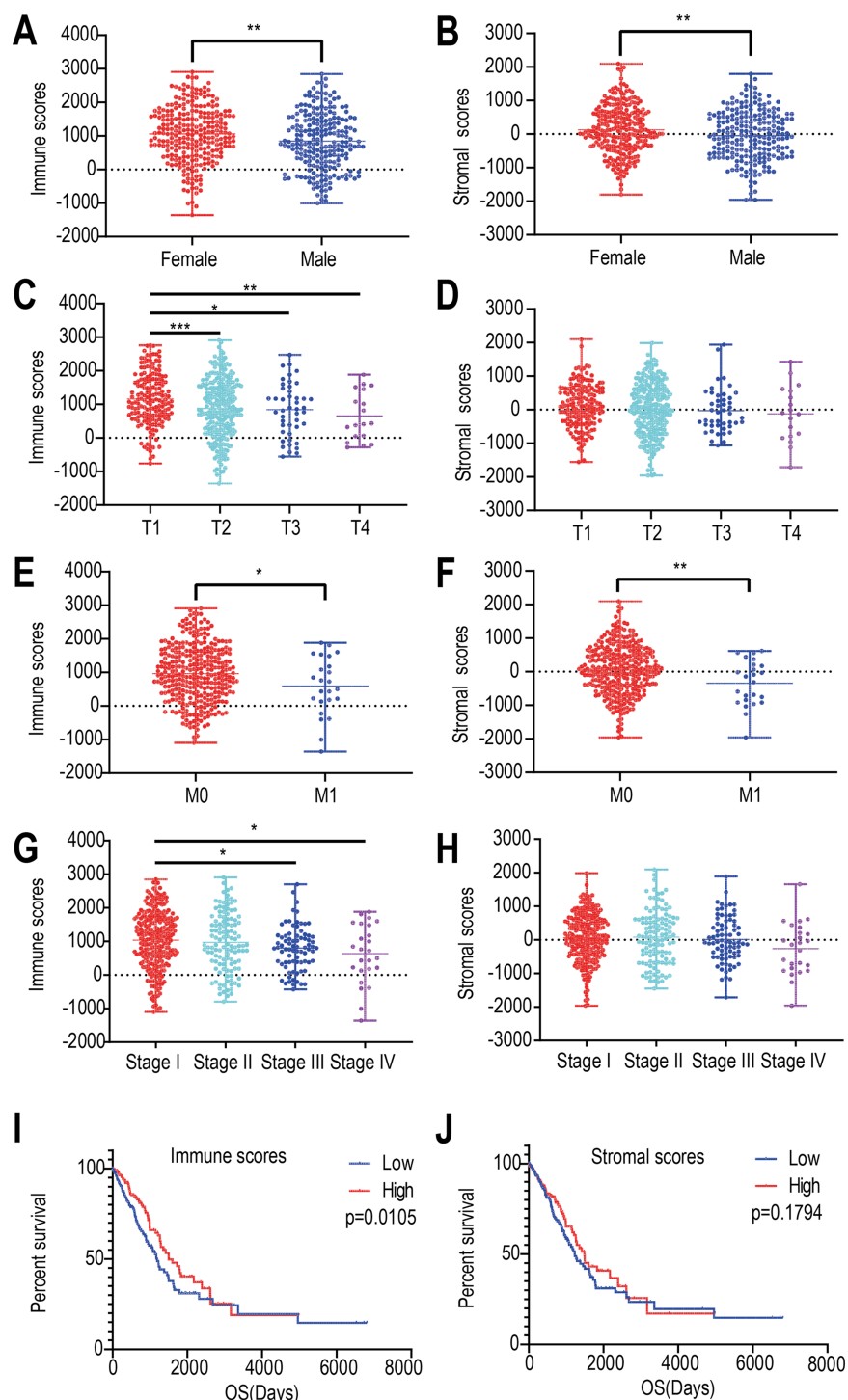

**Figure 1** **Relationships among immune and stromal scores, clinical characteristics, and overall survival.** Distribution of stromal and immune scores based on gender (A), T stage (B), M stage (C) , and pathological stage (D). Kaplan–Meier survival analysis of low and high score groups based on stromal (E) and immune scores (F). (* $p < 0.05$, ** $p < 0.01$, *** $p < 0.001$).

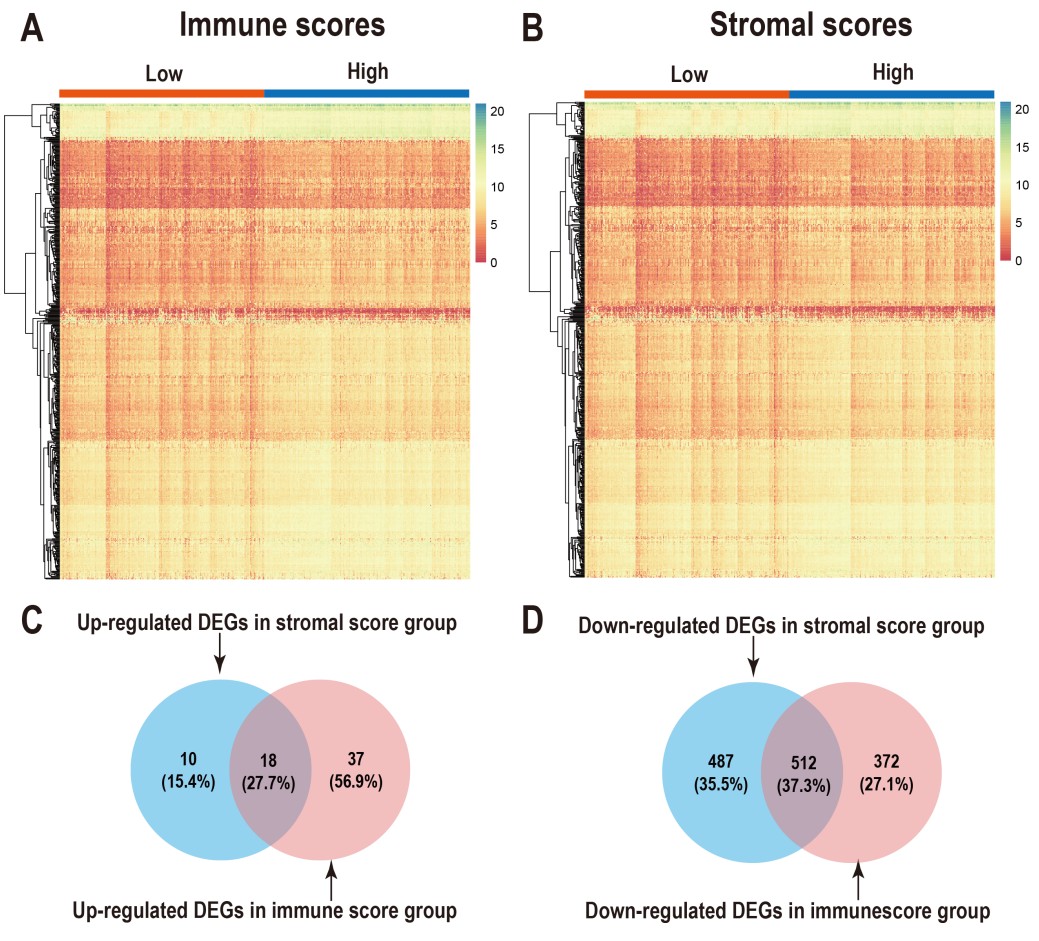

**Figure 2** **Comparison of gene expression profiles with immune scores and stromal scores in LUAD.**
Heatmaps were drawn for the DEGs obtained from the respective high and low group comparisons of the immune scores (A) and stromal scores (B). Venn diagrams drawn from the DEGs that are up- or down-regulated in immune scores (C) and stromal scores (D).

that of the high-score group (1235 d vs. 1492 d, $p = 0.179$; log-rank test; Fig. 1J), although it was not statistically significant.

## Comparison of gene expression profiles with LUAD immune scores and stromal scores

We performed differential expression analysis of the high- and low- immune scoring patients and found that 884 genes were downregulated and 55 genes were upregulated (Fig. 2A, Table S1). Similarly, differential analysis of the high- and low-stromal scoring groups revealed that 999 genes were downregulated and 28 genes were up-regulated ($|logFC|>1$, $p < 0.05$) (Fig. 2B, Table S1). In the Venn diagram, 18 genes were found to be upregulated in both groups (Fig. 2C) and 512 genes were downregulated in both groups (Fig. 2D).
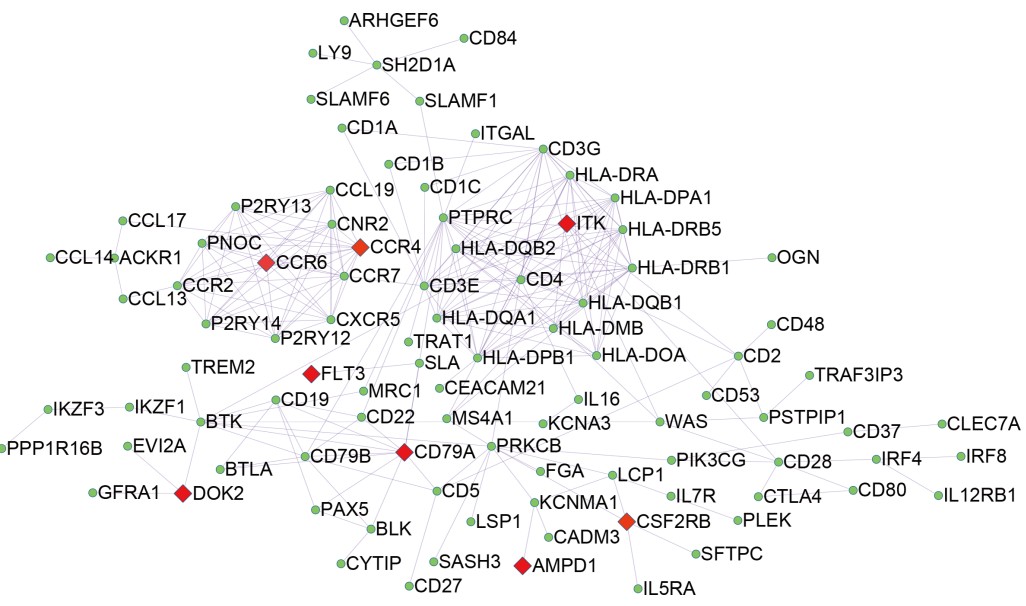

**Figure 3  PPI network of 286 prognostic DEGs.**

## Correlation of individual DEGs expression with overall survival and Functional enrichment analysis of prognostic associated genes

To explore the DEGs associated with overall survival, we performed a univariate Cox regression analysis. The results (Table S2) showed that 286 of the 530 DEGs were survival-related genes ($p < 0.05$). To further determine the relationship between these prognostic DEGs, we used the STRING online tool to obtain a PPI network. This network has 277 nodes and 2143 edges (Fig. 3). Functional enrichment clustering of 286 prognostic DEGs was also closely related to the immune pathways. GO analysis results showed that 46 terms were statistically significant, and KEGG enrichment results showed that 22 pathways were significantly enriched ($p < 0.05$, FDR < 0.05). The top GO terms included adaptive immune response, T cell co-stimulation, MHC class II receptor activity, and transmembrane signaling receptor activity (Fig. 4A). In addition, most pathways generated from the KEGG pathway enrichment analysis were associated with immune responses, such as cell adhesion molecules, primary immunodeficiency, antigen processing and presentation, and chemokine signaling pathways (Fig. 4B).

## Further identification of prognostic genes in the GEO database

To confirm whether the genes identified in the above steps have prognostic functions in other LUAD cohorts, we used the expression profiles and clinical data of three datasets in the GEO database for verification: GSE37745, GSE11969, and GSE50081. A total of 80 genes in GSE37745, 2703 genes in GSE11969, and 34 genes in GSE50081 were associated with survival. CSF2RB, inducible T cell kinase (ITK), and FLT3, and CD79A were found to be common among the GSE37745, GSE11969, and TCGA datasets (Fig. 5A). In addition, CCR4, CCR6, DOK2, AMPD1, and IGJ were found to be common among GSE37745, GSE50081, and TCGA (Fig. 5B). Kaplan–Meier survival curves were generated for the

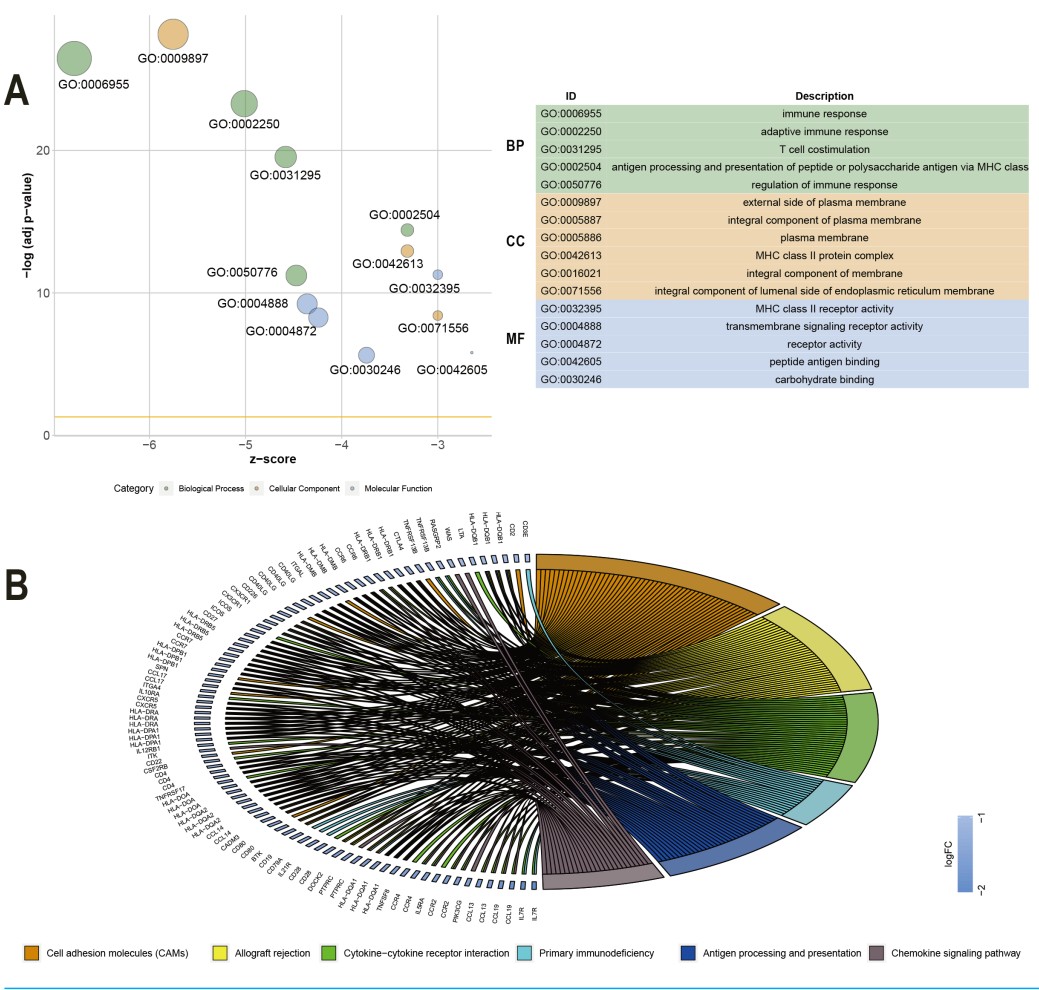

**Figure 4** **GO term and KEGG analysis of 286 DEGs.** (A) GO analysis of 286 DEGs. (B) KEGG enrichments of 286 DEGs. GO –gene ontology; KEGG –Kyoto Encyclopedia of Genes and Genomes; MF –molecular function; CC –cellular component; BP –biological process.

nine prognostic genes (Fig. 5C). The results showed that for all but except CCR4, the overall survival time of the low expression group was significantly lower than that of the high expression group (log-rank $p < 0.05$). This suggests that reduced expression of the eight genes (CSF2RB, ITK, FLT3, CD79A, CCR6, DOK2, AMPD1, and IGJ) and increased expression of one gene (CCR4) can predict poor survival of patients with for LUAD.

## Immune cell infiltration analysis revealed the correlation between the expression of the DEGs and immunocytes

To confirm these findings, we used multivariate Cox regression and found that the five genes, including CCR6, ITK, CCR4, DOK2, and AMPD1, were an independent prognostic indicator for LUAD in their specific data sets (Figs. 6A–6D). We investigated immune infiltration using the TIMER deconvolution approach. Interestingly, the expression of the five genes identified positively correlated to the infiltration level of different immune cells, including B cells, CD4$^+$ T cells, CD8$^+$ T cells, macrophage cells, neutrophil cells, and

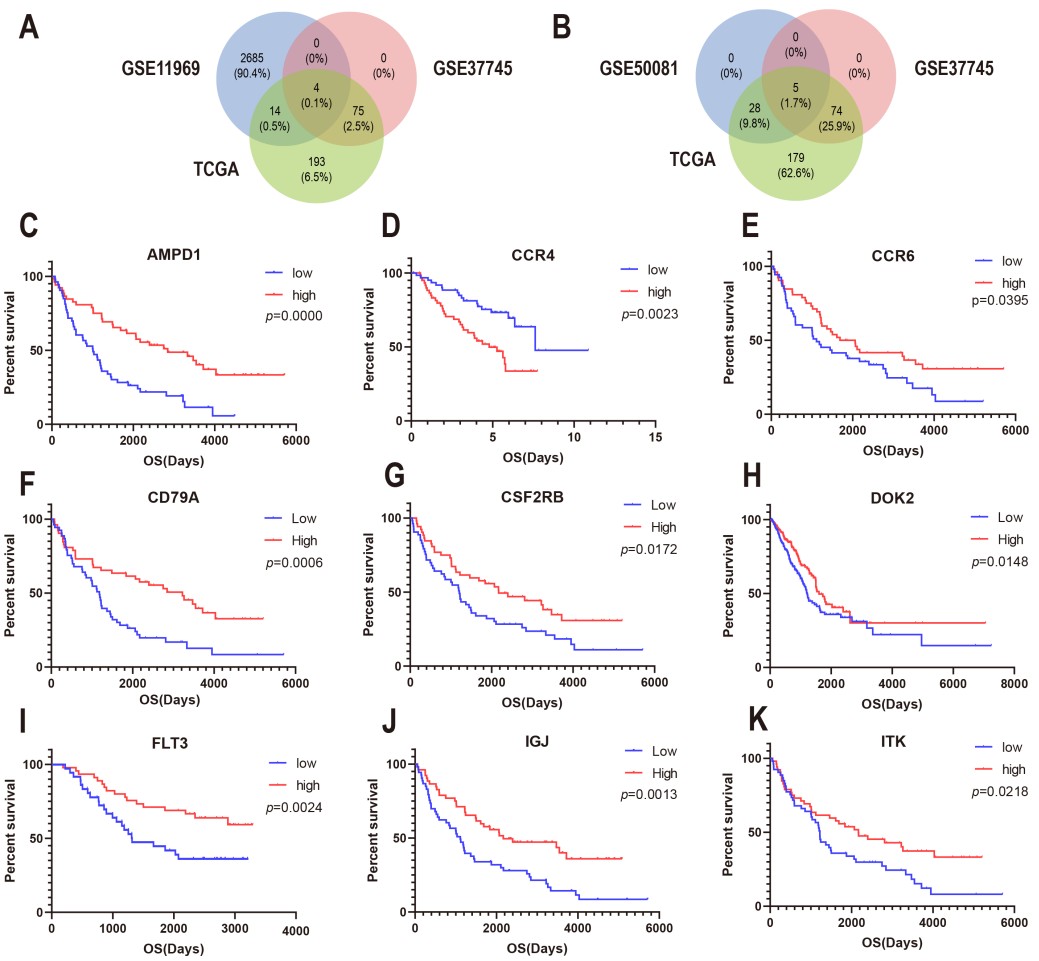

**Figure 5** Correlation between expression of nine prognosis-related genes and overall survival in GEO. (A) The Venn diagram shows prognosis-related genes based on GSE11969, GSE37745 and TCGA cross-validation. (B) The Venn diagram shows prognosis-related genes based on GSE50081, GSE37745, and TCGA cross-validation. (C) Kaplan–Meier survival curves were generated for nine prognosis genes (AMP11, CCR4, CCR6, CD79A, CSF2RB, DOK2, FLT3, IGJ, and ITK) extracted from the comparison of groups of high (red line) and low (blue line) gene expression.

dendritic cells ($p$ <0.05). The expression levels of the five genes were negatively correlated with the tumor purity ($p < 0.05$; Figs. 7A–7II).

## Evaluation of the immune and stromal status between the groups with low and high expression of CCR6

Multivariate Cox regression analysis identified that low expression of CCR6 is an independent prognostic factor of poor survival in the TCGA cohort of 484 patients with LUAD (HR 0.29, 95 % CI [0.12–0.72], $p = 0.007$). According to the results of UALCAN, mRNA and protein expression of CCR6 were all significantly upregulated in primary LUAD tissues as compared with normal samples (all $p < 0.05$; Figs. 8A and 8B). Immunohistochemistry staining obtained from The Human Protein Atlas database also

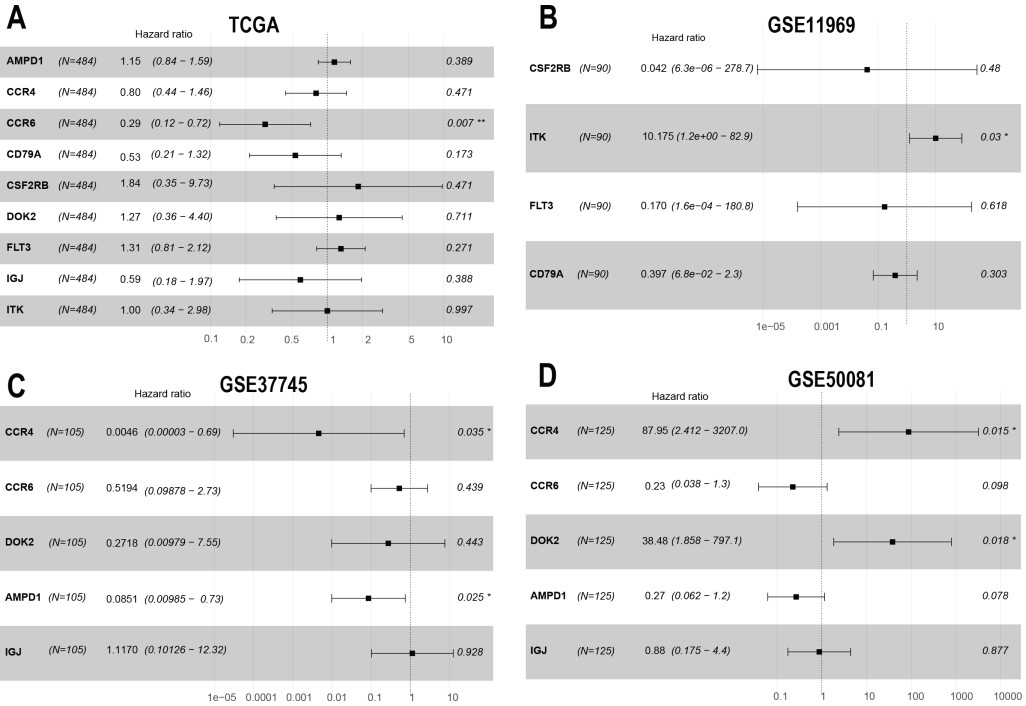

**Figure 6** **The multivariate Cox regression analysis of 9 prognostic genes.** The HR and *P*-value from the multivariate Cox regression of selected prognostic genes in TCGA (A), GSE11969 (B), GSE37745 (C), GSE50081 (D). (* $p < 0.05$, ** $p < 0.01$).

demonstrated that CCR6 was upregulated (Figs. 8C and 8D). According to the median value of CCR6 expression levels, we divided the 484 LUAD patients into a low expression group and a high expression group. CIBERSORT and xCell were used to depict the immune cell and stromal cell landscape of CCR6 with high and low expression levels. Figs. 8E and 8F show the proportions of immune cells and stromal cells in the 484 LUAD tissues. The differential proportion of infiltration of immune and stromal cells in the group with low CCR6 expression and the group with high CCR6 expression is shown in Figs. 8G and 8H. CIBERSORT analysis suggests that high CCR6 expression is associated with recruitment of memory B cells, resting memory CD4 T cells, monocytes, M1 macrophages, resting dendritic cells, and resting mast cells (Fig. S1). xCell analysis suggests that high CCR6 expression is associated with recruitment of adipocytes, chondrocytes, endothelial cells, and fibroblasts (Fig. S1). A comparison of the tumor infiltration levels among tumors with different somatic copy number alterations (SCNAs) for a given gene was determined using TIMER. The results showed that enrichment of the six immune cell types (B cell, CD4[+] T cell, CD8[+] T cell, macrophage cell, neutrophil cell, and dendritic cell) were significantly downregulated in LUAD with SCNA of CCR6 (Fig. 8I).

## DISCUSSION

Previous studies have shown that the TME can play a key role in promoting tumor progression and increasing mortality (*Caetano et al., 2016*; *Wood et al., 2014*). From the

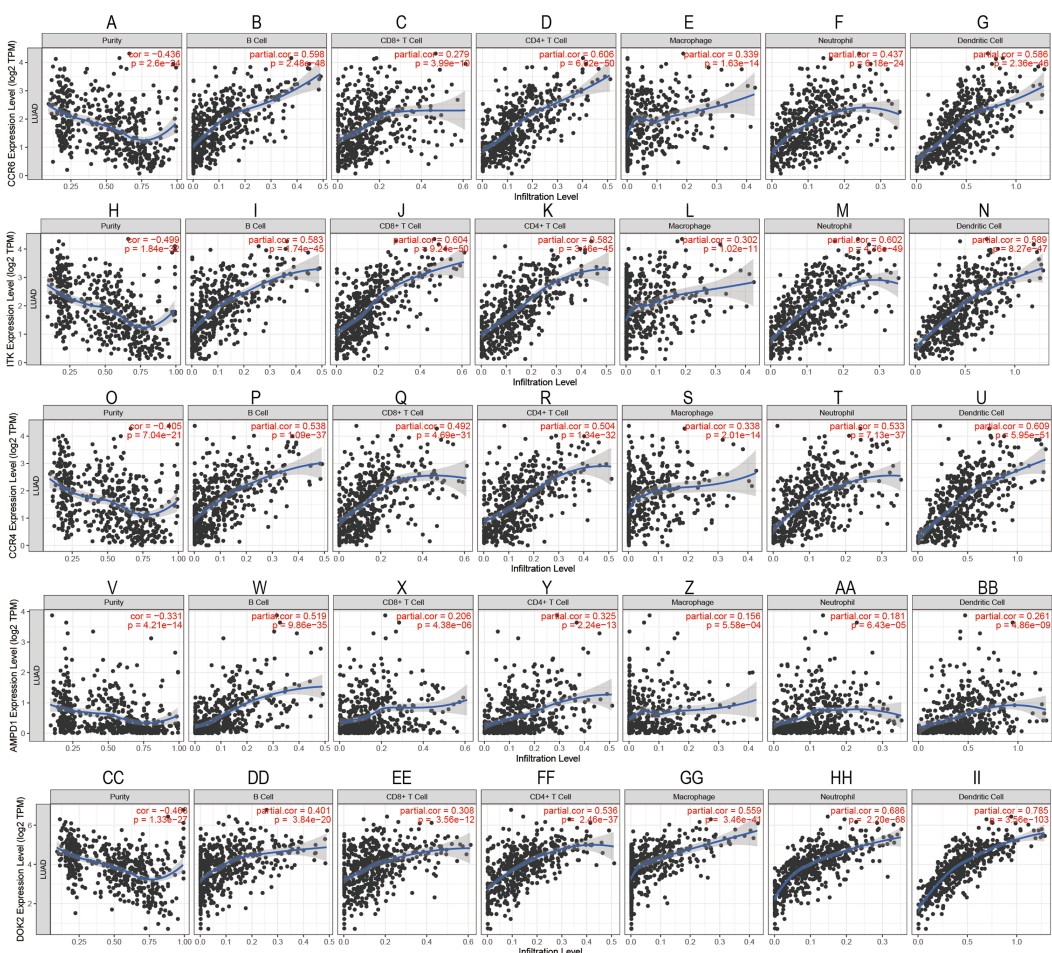

**Figure 7** Correlation analysis between the expressions of mainly identified DEGs (CCR6, ITK, CCR4, DOK2, AMPD1) and infiltration levels of B cell, CD8+ T cell, CD4+ T cell, macrophage, neutrophil, and dendritic cell.

perspective of the TME, we identified a series of genes related to the overall survival of patients with LUAD based on data in the TCGA database. By analyzing differences in the overall expression profiles of patients of LUAD cases in high and low scoring groups using the ESTIMATE algorithm, we identified 286 prognostic genes involved in immune response and chemokine signaling. Importantly, we validated these results using three independent LUAD datasets from the GEO database. From the overlap among these datasets, we obtained nine prognostic genes that were associated with the immune microenvironment of LUAD. We found that five (ITK, DOK2, AMPD1, CCR4, and CCR6) out of nine genes were prognostic factors for LUAD in their specific data set.

Abnormal expression of protein tyrosine kinases (PTK) is related to tumor invasion and metastasis, tumor neovascularization, and tumor chemotherapy resistance (*Knosel et al., 2014*). Many drugs have been developed to target PTKs. Although tyrosine kinase inhibitors (TKIs) continue to be involved in targeted therapies for lung cancer, acquired resistance still

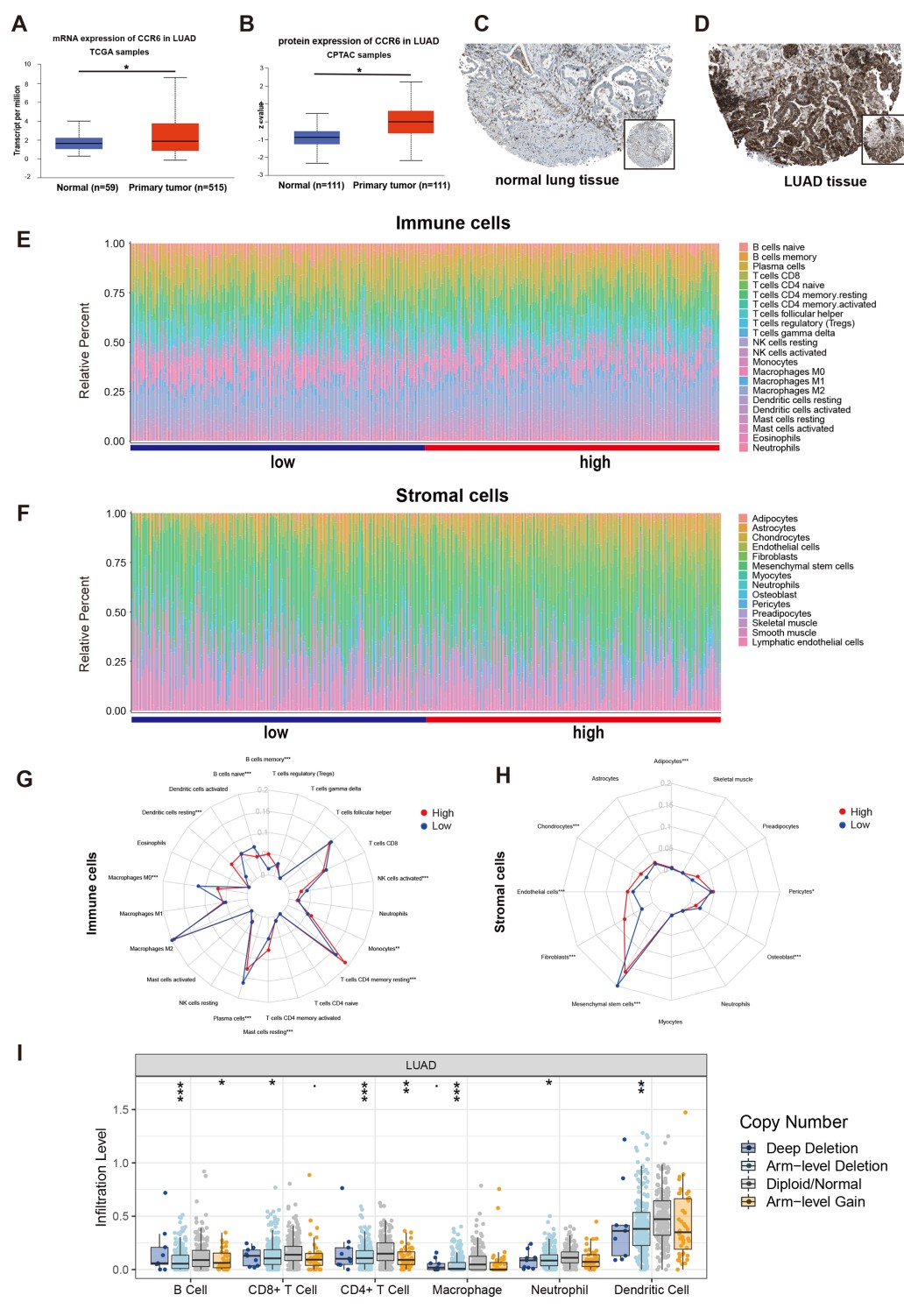

**Figure 8** **mRNA and protein expressions, immune and stromal status of CCR6 in LUAD.** (A) mRNA and protein expressions of CCR6 in primary LUAD tissues compared to normal samples (UALCAN). (B) CCR6 protein were not expressed in normal lung tissues, whereas its high expressions were observed in LUAD tissues (The Human Protein Atlas). (C) Bar charts of 22 immune cell proportions (CIBERSORT) and 14 stromal cell proportions (xCell) between low and high expression of CCR6 in LUAD. (D) Differential expression of different types of immune cells and stromal cells between low and high expression of CCR6 in LUAD. (E) The association of SCNAs of CCR6 with immune infiltration in LUAD (TIMER).

remains a problem (*Ahn et al., 2017*). It has been suggested that the mechanism of action of different PTKs needs additional study. IL2 inducible T cell kinase (ITK) is expressed in many immune cells. As a non-receptor tyrosine kinase, it plays an important role in regulating TCRs, chemokine receptor CXCR4, and FC $\varepsilon$R-mediated signaling pathways (*Sahu & August, 2009*). The proteins downstream of tyrosine kinase participate in the proliferation and migration of lung cancer (*Ghanem et al., 2014*), leukemia (*Tomoharu et al., 2004*), gastric cancer (*Liu & Xiao, 2014*), and other cancer cells via the receptor tyrosine kinase signaling pathway. A study has found that DOK2 is a tumor suppressor gene in LUAD and that knocking out Dok2 in mice accelerates lung tumorigenesis induced by oncogenic EGFR (*Berger et al., 2013*). We found that low expression levels of ITK, and DOK2 in LUAD were associated with poor prognosis, possibly affecting tumor progression via the transmembrane receptor PTK signaling pathway. However, further studies are required to determine whether TKI can become a new target for lung cancer.

The AMPD1 gene encodes adenosine monophosphate deaminase (AMPD), a crucial enzyme in purine nucleotide and energy metabolism, especially in skeletal muscle and cardiac muscle. The role of AMPD1 is to catalyze the conversion of adenosine monophosphate to inosine monophosphate (*Smolenski et al., 2014*). *Luo et al. (2018)* identified that AMPD1 was closely associated with the survival of breast cancer patients. *Zha & Wu (2018)* found that the expression of AMPD1 in serum of patients with papillary thyroid carcinoma is closely related to the malignant evolution of PTC and clinical prognosis of patients. The relationship between AMPD1 and LUAD has not previously been elucidated.

Functional enrichment analysis showed that CCR6 was significantly involved in the chemokine signaling pathway. This is consistent with results from previous studies that indicate that in the TME, the communication among cells is dynamically regulated by a complex network of immune factors, including cytokines, chemokines, and numerous growth factors (*Shimizu, Okita & Nakata, 2013*). A variety of chemokines promote tumor proliferation, which can play a vital role in tumor progression by helping tumor cells escape immune monitoring (*Chang et al., 2018*).

We found that low expression of CCR6 and high expression of CCR4 were significantly associated with poor prognosis of LUAD, based on a multi-cohort study. Multivariate Cox analysis suggests that CCR6 is an independent prognostic factor in the TCGA LUAD cohort and that CCR4 is an independent prognostic indicator in the two LUAD cohorts, GSE50081 and GSE37745. *Liu et al.(2017b)* also found that CCR4 is an independent risk factor for the overall survival of NSCLC, as it functions as a ligand for CCL17 and CCL22. CCR4 can help tumor cells escape the host immune attack by attracting Treg into the TME. Studies have identified that CCR6 can act as a ligand for CCL20 and promote lung metastasis of cancerous adrenal tissue (*Raynaud et al., 2010*). In a mouse model of lung cancer (Lewis lung carcinoma), expression of CCR6 in tumor cells reduced metastatic potential (*Sutherland et al., 2007*). Similar to our findings, a study found that increased expression of CCR6 in tumor cells is an independent predictor of a better prognosis in patients with LUAD (*Minamiya et al., 2011*). A recent study showed that CCL20, in coordination with CCR6, can recruit Treg cells to tumor sites (*Zhang et al., 2015*). To our

knowledge, we have for the first time depicted the landscape of immune cell infiltration and the landscape of CCR6 in stromal cells in LUAD. Our results also suggest that there is a close correlation between CCR6 genomic alterations and immune cell enrichments in LUAD. Therefore, we hypothesized that genetic alteration of CCR6 may play an important role in LUAD. Consistent with both of these previous studies, our analysis reveals an important role of CCR6 in LUAD and its potential value as a biomarker.

## CONCLUSIONS

In conclusion, using the ESTIMATE algorithm, TIMER, CIBERSORT, and xCell, we analyzed the expression profile data of patients with LUAD from the TCGA database and identified a series of genes related to the TME. Many previously unknown genes were found in addition to the genes already associated with lung cancer or immune microenvironments. Finally, we examined the interaction between CCR6 and TME. This may be important to identify the connecting mechanism between TME and LUAD prognosis.

### Funding

This study was supported by the National Natural Science Foundation of China (NSFC81660488) and the Guangxi Natural Science Foundation under Grant 2017GXNSFAA198123. The funders had no role in study design, data collection and analysis, decision to publish, or preparation of the manuscript.

### Grant Disclosures

The following grant information was disclosed by the authors:
The National Natural Science Foundation of China: NSFC81660488.
The Guangxi Natural Science Foundation: 2017GXNSFAA198123.

### Competing Interests

The authors declare there are no competing interests.

### Author Contributions

- Zhan-yu Xu and Mengli Zhao conceived and designed the experiments, performed the experiments, prepared figures and/or tables, and approved the final draft.
- Wenjie Chen and Fanglu Qin performed the experiments, prepared figures and/or tables, authored or reviewed drafts of the paper, and approved the final draft.
- Kun Li performed the experiments, prepared figures and/or tables, and approved the final draft.
- Wei-wei Xiang, Yu Sun and Jiangbo Wei performed the experiments, analyzed the data, authored or reviewed drafts of the paper, and approved the final draft.
- Li-qiang Yuan performed the experiments, analyzed the data, prepared figures and/or tables, and approved the final draft.
- Shi-kang Li and Sheng-hua Lin conceived and designed the experiments, performed the experiments, prepared figures and/or tables, and approved the final draft.

## Data Availability

Data are available at NCBI GEO: GSE37745, GSE11969, and GSE50081; and the TCGA database (Available at https://portal.gdc.cancer.gov/repository?facetTab=files&filters=%7B%22op%22%3A%22and%22%2C%22content%22%3A%5B%7B%22op%22%3A%22in%22%2C%22content%22%3A%7B%22field%22%3A%22cases.primary_site%22%2C%22value%22%3A%5B%22bronchus%20and%20lung%22%5D%7D%7D%2C%7B%22op%22%3A%22in%22%2C%22content%22%3A%7B%22field%22%3A%22cases.project.program.name%22%2C%22value%22%3A%5B%22TCGA%22%5D%7D%7D%2C%7B%22op%22%3A%22in%22%2C%22content%22%3A%7B%22field%22%3A%22cases.project.project_id%22%2C%22value%22%3A%5B%22TCGA-LUAD%22%5D%7D%7D%2C%7B%22op%22%3A%22in%22%2C%22content%22%3A%7B%22field%22%3A%22files.data_category%22%2C%22value%22%3A%5B%22transcriptome%20profiling%22%5D%7D%7D%2C%7B%22op%22%3A%22in%22%2C%22content%22%3A%7B%22field%22%3A%22files.experimental_strategy%22%2C%22value%22%3A%5B%22RNA-Seq%22%5D%7D%7D%5D%7D).

## Supplemental Information

Supplemental information for this article can be found online at http://dx.doi.org/10.7717/peerj.9530#supplemental-information.

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
