# Peer review of "Analysis of prognostic genes in the tumor microenvironment of lung adenocarcinoma"

_PeerJ, doi:10.7717/peerj.9530_

## Round 0.1 · original submission · Major Revisions

Manuscript which you submitted to PeerJ, has been reviewed. The reviewers have recommended publication pending major revisions. Therefore, I invite you to respond to the reviewers' comments at the bottom of this letter and revise your manuscript accordingly.

Reviewer 1 ·

Basic reporting

no comment

Experimental design

no comment

Validity of the findings

no comment

Additional comments

In this study, authors used bioinformatics to screen nine prognosis-related genes associated with the LUAD tumor microenvironment. However, the selected key genes need to do validation using Western blot or IHC staining in their collected cancer samples with paitients' prognosis, not just the data from datasets or website.Otherwise, the reliability of the results is not strong.

·

Basic reporting

In this manuscript, authors found nine prognosis-related genes associated with immune cells/stromal cells in LUAD using the ESTIMATE algorithm. Although the authors emphasized that the new biomarkers associated with immune cells/stromal cells played an important role in the prognosis of the LUAD, some contents are not fully abundant and controversial.
In terms of basic reporting, there is a problem that needs to be revised.In Line 110, the abbreviation ”logFC” lacked its full name.Other content basically meets the requirements.

Experimental design

With regard to the experimental design, the following content still needs to be added.
1.The standard and definition of the cutoff by which both the stromal scores and the immune scores were grouped should be provided in the full manuscript.
2.The results of univariate Cox regression mentioned in Line175-176 should be provided as supplementary material.

Validity of the findings

This manuscript is novel, reporting the new biomarkers based on the immune cells/stromal cells in LUAD using the ESTIMATE algorithm. However, the following results still needs to be modified.
1. In Line 35-37, it was mentioned that immune scores and stromal scores are significantly associated with the survival time. However, the results showed in the Line 154-155 and Figure 1E, didn’t indicated that there was a significant association between stromal scores and overall survival (p = 0.179). If the stromal scores didn’t significantly affect the prognosis in LUAD, is it accurate and necessary to conduct further research on it?
2. Were the immune scores and stromal scores the independent prognostic factors in LUAD? More information should be provided.
3. In Line 156-164, GSEA was performed to investigate mechanisms contributing to both the high immune and stromal scores. However, there was only a series of figures in the Figure 2, which wasn’t explicitly stated that the analysis is based on which score groups.
4. Because both the GSEA and DAVID analysis were performed using the GO analysis and KEGG pathway enrichment, consistency between the two results is inevitable. Thus, it is unknown whether DAVID analysis could actually verify the results in GSEA.
5. There were three GEO database for verification. However, the venn diagram showing prognosis-related genes based on GSE11969, GSE50081, GSE37745 and TCGA cross-validation was lacked. Please explain why only two database was chosen to cross-valid the TCGA datasets in Fig 6 A&B. Furthermore, the Kaplan–Meier survival analysis in Fig 6 suggested that the 9 genes might be the prognosis-related factors rather than independent ones. Further results should be provided.

Additional comments

See 'Validity of the findings' for details.

Reviewer 3 ·

Basic reporting

The authors have conducted an extensive study on the TME of LUAD. The manuscript was well written, with informative figures and tables and thorough results. The English language used is clear. The literature is well referenced and the conclusions based on the results are generally sound, however, citing some studies in the Introduction and Discussion sections would be beneficial and strengthen the manuscript.

In some instances, some changes should be taken into account to add consistency to the text:
1) Line 19-20, authors should write the identification of different prognostic genes in TME.
2) Line 30-31, How many DEGs were used to construct PPI.
3) Line 40, Which nine genes were found, please input the name of these genes.
4) Line 49-50, Eighty-five percent of lung cancers are non-small cell lung......., references should be included.
5) Line 59-60, Please clarify the sentence.
6) Line 183, replace “value” with “associated”.
7) In figure1A-D, please mention the statistical threshold in legend and please mention the statistical threshold in all legends of figures.
8) In figure 1E, please clearly indicate the meaning of all different lines in survival curve.
9) In figure 3C-D, title of circle should be more clear. Title of circle is confusing.
10) In figure 6C, clearly indicate the meaning of all different lines in survival curve because there are many line and it is confusing.
11) The legends of every figure should be more descriptive, clear, and cosistent with the diagram.

Experimental design

Databases and analyses used were generally adequate to reach the objectives, however, some suggestions should be taken into account for the robustness of this study:
1) Line 94, the website should be mentioned.
2) Line 95-96, the database number should be provided.
3) Line 180-181, the name of some important genes should be given.
4) In Introduction, add some text with references indicating that stromal genes are associated with the poor survival of patients in different cancers, including lung cancer.
5) Because stroma encompasses many type of cells, to improve the quality of the manuscript, I suggest the authors to search the GEO database on stroma of LUAD and validate the expression and prognostic value of genes.
6) The authors can identify the prognostic hub genes from PPI networks which may improve the quality of this study.
7) In figure 5A, provide the significant p values for each GO.

Validity of the findings

To prove the validity of the findings, the authors should search the databases on single stroma or immune cells of lung cancer to identify which part of the stromal compartments or immune cells the prognostic nine genes are associated with.

Additional comments

The findings of this study are potentially useful for establishing appropriate management and treatment of lung cancer. The conclusions could be improved with a more detailed discussion of the background and a more cautious interpretation of results.

Reviewer 4 ·

Basic reporting

There were so many grammar mistakes, so please edit the manuscript by a native English speaker.

Experimental design

The DEGs in high immune and stromal score group from the TCGA database should be provided in the supplementary files.

Validity of the findings

The prognostic accuracy of immune and stromal score should also be validated using cases from the GEO database. You just analyzed the intersect genes between cases from the GEO and TCGA databases, it was not the validation of the findings from the TCGA database.

Additional comments

1. Too many categories of p-value in Table 1, I think just p<0.05* and 0.01** are enough.
2. A prognostic model should be constructed based on the DEGs, which can be identified through LASSO regression.

---

## Round 0.2 · Minor Revisions

The manuscript "Analysis of prognostic genes in the tumor microenvironment of lung adenocarcinoma " which you submitted to PeerJ, has been peer-reviewed. I invite you to respond to the reviewer's comments at the bottom of this letter and revise your manuscript accordingly.

Reviewer 4 ·

Basic reporting

There were so many grammar mistakes, so please edit the manuscript by a native English speaker. Such as “In addition to tumor cells, stromal cells, immune cells, and tumor-associated normal epithelium…” in line 65.

Experimental design

Please figure out which parameters were analyzed with student’s t-test and one-way statistics and which statistical software was used.

Validity of the findings

Just analyze the distribution of DEGs included in both immune and stromal score group in three GEO datasets.

Additional comments

1. Too many categories of p-value in Table 1, I think just three decimal places of p-value are enough.

---

## Round 0.3 · accepted · Accept

The authors have greatly improved their manuscript, which can be accepted now.